# Design, Synthesis, Molecular Modeling, and Anticancer Evaluation of New VEGFR-2 Inhibitors Based on the Indolin-2-One Scaffold

**DOI:** 10.3390/ph15111416

**Published:** 2022-11-15

**Authors:** Mohamed A. Abdelgawad, Alaa M. Hayallah, Syed Nasir Abbas Bukhari, Arafa Musa, Mohammed Elmowafy, Hamdy M. Abdel-Rahman, Mohammed K. Abd El-Gaber

**Affiliations:** 1Department of Pharmaceutical Chemistry, College of Pharmacy, Jouf University, Aljouf 72341, Saudi Arabia; 2Pharmaceutical Organic Chemistry Department, Faculty of Pharmacy, Assiut University, Assiut 71526, Egypt; 3Pharmaceutical Chemistry Department, Faculty of Pharmacy, Sphinx University, New Assiut 71515, Egypt; 4Department of Pharmacognosy, College of Pharmacy, Jouf University, Aljouf 72341, Saudi Arabia; 5Department of Pharmaceutics, College of Pharmacy, Jouf University, Aljouf 72341, Saudi Arabia; 6Medicinal Chemistry Department, Faculty of Pharmacy, Assiut University, Assiut 71526, Egypt; 7Pharmaceutical Chemistry Department, Faculty of Pharmacy, Badr University, Assiut 2014101, Egypt

**Keywords:** anti-proliferative, apoptosis, indolin-2-one, isatin, docking, sunitinib, VEGFR-2 inhibitors

## Abstract

A new series of indoline-2-one derivatives was designed and synthesized based on the essential pharmacophoric features of VEGFR-2 inhibitors. Anti-proliferative activities were assessed for all derivatives against breast (MCF-7) and liver (HepG2) cancer cell lines, using sunitinib as a reference agent. The most potent anti-proliferative derivatives were evaluated for their VEGFR-2 inhibition activity. The effects of the most potent inhibitor, **17a**, on cell cycle, apoptosis, and expression of apoptotic markers (caspase-3&-9, BAX, and Bcl-2) were studied. Molecular modeling studies, such as docking simulations, physicochemical properties prediction, and pharmacokinetic profiling were performed. The results revealed that derivatives **5b**, **10e**, **10g**, **15a**, and **17a** exhibited potent anticancer activities with IC_50_ values from 0.74–4.62 µM against MCF-7 cell line (sunitinib IC_50_ = 4.77 µM) and from 1.13–8.81 µM against HepG2 cell line (sunitinib IC_50_ = 2.23 µM). Furthermore, these compounds displayed potent VEGFR-2 inhibitory activities with IC_50_ values of 0.160, 0.358, 0.087, 0.180, and 0.078 µM, respectively (sunitinib IC_50_ = 0.139 µM). Cell cycle analysis demonstrated the ability of **17a** to induce a cell cycle arrest of the HepG2 cells at the S phase and increase the total apoptosis by 3.5-fold. Moreover, **17a** upregulated the expression levels of apoptotic markers caspase-3 and -9 by 6.9-fold and 3.7-fold, respectively. In addition, **17a** increased the expression level of BAX by 2.7-fold while decreasing the expression level of Bcl-2 by 1.9-fold. The molecular docking simulations displayed enhanced binding interactions and similar placement as sunitinib inside the active pocket of VEGFR-2. The molecular modeling calculations showed that all the test compounds were in accordance with Lipinski and Veber rules for oral bioavailability and had promising drug-likeness behavior.

## 1. Introduction

The treatment of cancerous diseases represents an exceptionally challenging battle for medicinal chemists to develop potent and safe chemotherapeutic agents [1,2,3,4]. These efforts mainly aim to identify and target the various biochemical processes involved in the progression and metastasis of tumors [5,6]. Angiogenesis, a complex process that involves the growth of new blood vessels from preexisting vasculature, is essential for normal organ growth and wound healing [7,8]. However, its imbalance is involved in the pathogenesis of different disorders, including cancer, psoriasis, multiple sclerosis, diabetic neuropathy, and rheumatoid arthritis [9,10,11,12,13].

Angiogenesis plays a critical role in tumor growth, invasion, and metastasis. Solid tumors are in dire need of new blood capillaries for an adequate supply of nutrients, the elimination of metabolic waste, and metastatic growth beyond their critical size [14].

The stimulation of various pre-angiogenic factors initiates angiogenesis through multiple steps, including basement membrane dissolution, migration, proliferation of endothelial cells, and finally, the formation of new capillary vessels [15,16].

Angiogenesis is controlled by several angiogenic regulators, such as vascular endothelial growth factor (VEGF) [17], fibroblast growth factor (FGF) [18], platelet-derived growth factor (PDGF) [19], transforming growth factor–β (TGF-β) [20], matrix metalloproteinases (MMPs) [21], epidermal growth factor (EGF) [22], angiopoietins [23], and integrins [24]. Among them, VEGF stands out as the most critical regulator of tumor angiogenesis [25,26]. Its actions are mainly mediated through a specific tyrosine protein kinase receptor called vascular endothelial growth factor receptor-2 (VEGFR-2) [27,28].

The binding of VEGF to VEGFR-2 leads to dimerization of two monomeric receptors and autophosphorylation of the tyrosine residues at the tail of the receptor’s intracellular domain, which initiates a signal transduction cascade that activates downstream signaling pathways [29]. These actions ultimately lead to angiogenesis and stimulate microvascular permeability, tumor proliferation, and tumor migration [30]. VEGFR-2 is overexpressed or hyperactivated in several kinds of malignancies, such as ovarian cancer [31], thyroid cancer [32], breast cancer [33], renal cancer [34], and hepatocellular carcinoma [35]. Therefore, blocking VEGF/VEGFR-2 signaling has been considered one of the most promising strategies for inhibition of angiogenesis and stopping tumor progression [36].

Over the past decades, many potent VEGFR-2 inhibitors have been approved by the Food and Drug Administration (FDA) to treat different types of tumors (Figure 1). According to the binding mode, VEGFR-2 inhibitors are divided into three major types [37]. Type I inhibitors are fitted in the conservative ATP active site and interact with Glu917 and Cys919 residues in the hinge region by hydrogen bonds [37]. Type I inhibitors, such as sunitinib and nintedanib, mostly bind to the active (DFG-in) conformation of VEGFR-2 [37,38]. Meanwhile, type II inhibitors target the inactive (DFG-out) conformation of VEGFR-2; these inhibitors bind to both the ATP catalytic site and the adjacent hydrophobic pocket [39]. Sorafenib, cabozantinib, lenvatinib, and tivozanib are examples of type II inhibitors [39,40]. Type III inhibitors are covalent noncompetitive inhibitors; these inhibitors bind through a covalent bond with a cysteine amino acid residue and prevent the binding of ATP at the binding site [38]. Vatalanib is considered an example of a type III inhibitor [38,41].

### Rationale and Design of the Work

The reported virtual screening and pharmacophore modeling studies revealed that most VEGFR-2 inhibitors contain four essential pharmacophoric features [42,43,44,45]. The reported features are: (1) a flat heteroaromatic ring contains hydrogen bond acceptor center that interacts with the key Glu917 and Cys919 in the ATP binding domain (hinge region) through hydrogen bond formation [43]; (2) a central aromatic system that occupies the linker region [46]; (3) a hydrogen bond acceptor and hydrogen bond donor moiety (HBA-HBD) that fits into the DFG domain [47]; (4) a terminal hydrophobic tail placed inside the hydrophobic allosteric pocket (Figure 2) [48,49].

According to the previous findings and based on our previous work on anticancer agents generally and kinase inhibitors specifically [50,51,52], we reported the design of a new series of VEGFR-2 inhibitors based on the indoline-2-one scaffold [43] to attain more potent anticancer agents. Different bioisosteric modifications, including replacement, extension, ring closure, and expansion, were applied at the four essential pharmacophoric features to generate our candidate compounds, as illustrated in (Figure 3).

The privileged indolin-2-one scaffold is regarded as one of the most promising heteroaromatic pharmacophore moieties that bind to the hinge region in the ATP active pocket of VEGFR-2 [53,54]. Therefore, the indolin-2-one nucleus was selected as the heterocyclic aromatic ring system for our target compounds. Secondly, the phenylimino moiety was chosen as the central aromatic system that occupies the linker region. Such a linker was selected to extend the available aromatic surface for interaction through ring expansion of the pyrrole ring of sunitinib to benzene and offer a possible new site for interaction through the imino nitrogen atom.

The third pharmacophoric HBA-HBD feature was realized using various moieties; ester (compounds **7a**,**b**), thiosemicarbazide (compounds **5a**,**b** and **17a**,**b**), hydrazide (compounds **10a**–**g** and **14a**,**b**), *N*-(2,5-dioxopyrrolidin-1-yl)amide (compounds **12a**,**b**), and oxadiazole ring (compounds **15a**,**b**). Different HBA-HBD moieties were chosen to offer a wide selection of hydrogen-bond-rich groups with different geometries and variable metabolic stabilities. Lastly, the terminal hydrophobic tail was varied to be either aliphatic chains (compounds **5a**,**b**, **7a**,**b**, and **14a**,**b**) or aromatic heterocyclic rings (compounds **12a**,**b**, and **15a**,**b**), or a substituted benzene ring with different hydrophobic, electronic, and topological groups (compounds **10a**–**g** and **17a**,**b**).

Therefore, in the current study and guided by preliminary docking studies, a new series of VEGFR-2 inhibitors based on the indolin-2-one scaffold was designed and synthesized in an endeavor to obtain potent anticancer agents. The growth inhibition activities for all the prepared compounds against the MCF-7 and HepG2 cancer cell lines were evaluated using sunitinib as the reference agent. The most potent anti-proliferative derivatives were tested for their VEGFR-2 inhibition activity. The molecular docking simulations were accomplished to predict the affinity and binding properties with VEGFR-2. Further biological investigations for the most potent inhibitor, **17a**, on cell cycle, apoptosis, and expression of caspase-3&-9, BAX, and Bcl-2, were assessed to gain a better understanding of its apoptotic activity. Finally, in silico physicochemical properties and pharmacokinetic profiling were calculated to ensure the drug-likeness ability of the designed compounds.

## 2. Results and Discussion

### 2.1. Chemistry

The preparation methodologies adopted to synthesize the target compounds **3a**,**b**–**17a**,**b** are outlined in Figure 1, Figure 2 and Figure 3. The structures of the final compounds were supported by various spectral and elemental analyses. The final compounds were obtained as a mixture of *E* and *Z* isomers, and the spectral data were reported for the major isomer.

The synthesis of the designed *N*-methylthiosemicarbazides **5a**,**b** is outlined in Figure 1. First, benzoic acid derivatives **3a**,**b** were obtained by condensing isatins **1a**,**b** with 4-aminobenzoic acid **2** in refluxing ethanol and catalytic glacial acetic acid [55]. Next, acids **3a**,**b** were subjected to an EDC/HOBt assisted coupling with 4-methylthiosemicarbazide **4** to afford the desired *N*-methylthiosemicarbazides **5a**,**b** in 78–82 % yield [56].

The preparation of the target benzhydrazidehydrazones **10a**–**g** is described in Figure 2. The intermediate Schiff base esters **7a**,**b** were prepared by condensation of isatins **1a**,**b** with benzocaine **6** in ethanol under acetic acid catalysis [57]. The hydrazinolysis of esters **7a**,**b** with hydrazine afforded the benzohydrazide derivatives **8a**,**b** in excellent 92–93% yield. The target hydrazones **10a**–**g** were prepared by condensation of hydrazides **8a**,**b** with different benzaldehyde derivatives **9a**–**g** using catalytic glacial acetic acid [58]. 

The synthetic procedures followed for preparation of the target compounds **12a**,**b**–**17a**,**b** are shown in Figure 3. The cyclic imides **12a**,**b** were synthesized by cyclodehydration of hydrazides **8a**,**b** with phthalic anhydride **11** in glacial acetic acid under sonication at 50 °C [59]. Treatment of hydrazides **8a**,**b** with ethyl acetoacetate **13** under the acetic acid catalysis afforded esters **14a**,**b** in excellent 86–88% yield [60]. 

Next, hydrazides **8a**,**b** were subjected to cyclization conditions with carbon disulfide and potassium hydroxide followed by acidification to furnish the oxadiazole derivatives **15a**,**b** in 82–86% yield [61]. The *N*-phenylthiosemicarbazides **17a**,**b** were prepared by reaction of benzhydrazides **8a**,**b** with phenyl isothiocyanate **16** in refluxing ethanol for eight hours. 

### 2.2. Biological Investigation

#### 2.2.1. In Vitro Cytotoxic Activity Assay

The cytotoxic activities against breast MCF-7 and liver HepG2 cancer cell lines for all the synthesized compounds **3a**,**b**–**17a**,**b** were evaluated by MTT-based cytotoxicity assay [62]. Sunitinib was selected as a reference standard in this study, and the results are presented as IC_50_ in Table 1.

The obtained data revealed that the tested derivatives, **3a**,**b**–**17a**,**b,** exhibited moderate to potent cytotoxicity against the MCF-7 (IC_50_ = 0.74–78.40 μM) and HepG2 (IC_50_ = 1.13–130.20 μM) cell lines compared to the reference sunitinib. Compound **10g** (IC_50_ = 0.74 ± 0.03 μM) displayed the most potent anti-proliferative activity against the MCF-7 cell line that was 6-folds more potent than sunitinib. Compound **5b** also inhibited the growth of the breast MCF-7 cancer cell line with (IC_50_ = 0.99±0.04 μM), which was 5-folds more potent than sunitinib. In addition, **10g** and **5b** displayed a comparable cytotoxic activity to sunitinib against the hepatic HepG2 cell line. Compound **17a**, compared with sunitinib, exhibited three-fold more potent growth inhibition activity against the MCF-7 cells and two-fold more potent cytotoxicity against the HepG2 cells (IC_50_ values of 1.44 ± 0.11 and 1.133 ± 0.06 μM), respectively. Moreover, compound **15a** successfully inhibited the growth of the MCF-7 cells (IC_50_ = 2.77 ± 0.10, 1.7-fold more potent than sunitinib) and the HepG2 cell line (IC_50_ = 2.303 ± 0.18 μM). Furthermore, hydrazone **10e** showed potent growth inhibition for the MCF-7 and HepG2 cell lines, with the MCF-7 cell line two times more sensitive than the HepG2 cell line. 

Generally, the fluorinated derivatives showed better cytotoxic activity than their unsubstituted counterparts, except for **8b**, **14b**, **15b**, and **17b** derivatives. Remarkably, it was observed that the lack of hydrophobic tail pharmacophoric feature in compounds **3a**,**b** and **8a**,**b** resulted in a significant drop in the cytotoxic activity, which clearly indicates the importance of this feature for efficient binding with the allosteric hydrophobic pocket within VEGFR-2. Additionally, among the series of benzohydrazidehydrazones **10a**–**g**, compound **10g**, which has the highest iLOG *P* (2.93), emerged as the most potent derivative against the MCF-7, which demonstrates the importance of hydrophobic interactions and the efficacy of the hydrazide group as the HBA-HBD moiety for VEGFR-2 inhibitory activity. The use of bulky cyclic phthalimide moiety as the hydrophobic tail in compounds **12a**,**b** decreased the cytotoxic activity, which may be attributed to steric bulkiness and inability of the HBA-HBD group to freely rotate for efficient binding inside VEGFR-2. The cytotoxicity test results were promising to evaluate the most potent derivatives for in vitro VEGFR-2 inhibition.

Incorporating the pharmacophoric HBA-HBD moiety within an oxadiazole ring resulted in derivatives **15a**,**b**, which showed promising cytotoxic activities. Oxadiazole **15a** was two times more potent than sunitinib against the MCF-7 cell line and exhibited close potency against the HepG2 cell line. Additionally, utilizing the thiosemicarbazide group as the HBA-HBD moiety proved to be very efficient for cytotoxic activity. The *N*-methyl derivative **5b** was the second most potent against the MCF-7 cell line, and the *N*-phenyl derivative **17a** was the most potent against the HepG2 cell line. 

#### 2.2.2. In Vitro VEGFR-2 Kinase Inhibitory Assay

The most potent antitumor derivatives, **5b**, **10e**, **10g**, **15a**, and **17a,** were tested for VEGFR-2 kinase inhibition using sunitinib as the reference compound; the results are reported as IC_50_ in Table 2. [63]. 

The outcomes showed that the tested compounds displayed potent inhibitory activities with IC_50_ from 0.078–0.358 μM. Compound **17a** demonstrated the most potent VEGFR-2 inhibition activity that was 1.78-fold more potent than sunitinib (IC_50_ = 0.078 and 0.139 μM, respectively). Moreover, the inhibitory activity of compound **10g** was 1.6-fold more than sunitinib (IC_50_ = 0.087 ± 0.004 μM). Derivatives **5b**, **10e**, and **15a** demonstrated promising VEGFR-2 inhibition with IC_50_ values that were 0.160 ± 0.008, 0.358 ± 0.019, and 0.180 ± 0.009, respectively. 

#### 2.2.3. Cell Cycle Analysis

Compound **17a** showed potent cytotoxic and VEGFR-2 inhibition activities, and was selected to explore its activity on the cell cycle distribution and cell proliferation of HepG2 cells. The HepG2 cells were exposed to **17a** (1.13 µM equal to its anti-proliferative IC_50_) for 24 h, and cell cycle progression was monitored by flow cytometry; the results are reported in Table 3 [64]. 

The obtained data revealed that compound **17a** decreased the distribution at the G0-G1 phase (42.91%) and the G2/M phase (9.07%) compared with the control (49.02 and 17.31%, respectively). In addition, the percentage of cell population increased at the S phase by 1.43-fold more than the control. These findings revealed that compound **17a** induced arrest of the cell cycle of the HepG2 cells at the S phase (Figure 4). Sorafenib was reported to induce a cell cycle arrest at the S phase and G2/M phase in HepG2 liver cancer cells [65].

#### 2.2.4. Apoptosis Analysis

The HepG2 cells were treated with **17a** (1.13 µM for 24 h), and the apoptotic effect was determined using Annexin V-FITC/PI assay. The results (Table 4) demonstrated that **17a** enhanced total apoptosis by 24-fold compared to the control (46.38% and 1.91%, respectively). Additionally, **17a** increased the percentage of early apoptosis compared with the control HepG2 cells (33.86% and 0.63%, respectively). Moreover, it increased the percentage of late apoptotic cells by 74-fold more than the control cells (from 0.17% to 7.99%). In addition, **17a** enhanced the necrosis percentage 4-fold more than the control. These details suggested that compound **17a** could induce the apoptotic mechanism of programed cell death in the HepG2 cell line (Figure 5).

#### 2.2.5. Caspase-3 and -9 Expression Assay

The expression of the apoptotic markers (caspase-3&-9) in the HepG2 cells treated with **17a** was studied to investigate the signal transduction pathway for its apoptotic activity.

The gene expression fold change of caspase-3 and -9 in the HepG2 cells treated with 1.13 µM of compound **17a** for 24 h was determined using quantitative real-time PCR analysis. The obtained results (Table 5) revealed that **17a** elevated the gene expression of caspase-3 by 6.9-fold and caspase-9 by 3.7-fold more than the control HepG2 cells. The obtained data suggested that the caspase transduction pathway is involved in the apoptotic effect of compound **17a** (Figure 6).

#### 2.2.6. BAX and Bcl-2 Expression Assay

The apoptotic BAX and anti-apoptotic Bcl-2 proteins play critical roles in caspase-independent apoptosis [66]. The ratio of the two proteins indicates the liability of a cell to be subjected to mitochondrial apoptosis [67]. 

The expression of BAX and Bcl-2 were evaluated in the HepG2 cells after treatment with 1.13 µM of compound **17a** for 24 h. The Western blotting technique was utilized to determine the levels of Bax and Bcl-2 proteins and estimate the Bax/Bcl-2 ratio.

The results (Table 6) showed that **17a** increased the expression of BAX by 2.7-fold more than the control cells. Moreover, it exhibited a pronounced decline in the expression level of Bcl-2 by 1.9-fold in comparison with the control. Additionally, compound **17a** enhanced the BAX/Bcl-2 ratio by 5-fold. These outcomes signified that compound **17a** induced mitochondrial apoptosis in the HepG2 cells (Figure 7).

### 2.3. Molecular Modeling Studies

#### 2.3.1. Docking Study

Molecular docking is the most utilized virtual drug design technique when the 3D structure of the target protein is available [68]. Simulations were carried out to predict the affinity and investigate the potential binding patterns of derivatives **3a**,**b**–**17a**,**b** within the active pocket of VEGFR-2. A docking study was performed on VEGFR-2 tyrosine kinase co-crystallized with sunitinib (PDB: 4AGD) [69] using MOE 2020.09 computational software [70]. First, validation of the molecular docking protocol was established by re-docking sunitinib in the ATP binding domain of the VEGFR-2 active pocket. Reproduction of the same binding interactions and orientation inside the active site as the co-crystallized ligand demonstrated that the applied docking setup was appropriate for the study. This was also confirmed by the small RMSD obtained (0.6797 Å) between the native ligand and the re-docked one.

Sunitinib achieved a docking score of −16.5974 kcal/mol and was interacted by the NH and C=O groups of its indolin-2-one with the Glu917 and Cys919 of hinge residues, respectively. Additionally, it showed multiple hydrophobic interactions with Leu840, Ala866, Val916, Phe918, and Leu1035 (Figure 8).

The docking simulation of **5b** in the ATP binding pocket of VEGFR-2 (Figure 9) showed that **5b** was fitted in the hinge region and the docking score was −19.0360 kcal/mol. Compound **5b** formed two hydrogen bond interactions by the NH and CO groups of its indolin-2-one scaffold with Glu917 and Cys919 hinge residues, respectively. Additionally, several hydrophobic and Van der Waals interactions were observed with Leu840, Val848, Ala866, Val916, Phe918, Leu1035, and Phe 1047.

The docking pose of compound **10g** into the ATP binding domain of VEGFR-2 (Figure 9) showed two hydrogen bond interactions: isatin NH of the ligand with Glu917 and isatin C=O with Cys919. Additionally, several Van der Waals and hydrophobic contacts were observed with Val848, Ala866, Val916, Phe918, Leu1035, and Phe 1047. These interactions were reflected in the docking score of **10g** (−21.2368 kcal/mol) and supported the obtained in vitro activity results.

The binding mode of compound **15a** (docking score −17.2599 kcal/mol) inside VEGFR-2 is shown in Figure 10. It displayed two hydrogen bond interactions by the NH and C=O groups of its isatin ring with Glu917 and Cys919 residues, respectively. Furthermore, a dipole interaction was observed between the S atom of the oxadiazole ring and Lys920 residue. Moreover, various Van der Waals and hydrophobic contacts were observed with Leu840, Val848, Ala866, Val916, Phe918, Leu1035, and Phe 1047.

Considering the binding mode of compound **17a** into the active pocket of VEGFR-2 (docking score −20.1061 kcal/mol), it formed two hydrogen bonds by the NH and C=O groups of its indolinone nucleus with Glu917 and Cys919, respectively (Figure 10). Additionally, an arene cation interaction was noticed between the terminal benzene ring and Lys838 residue. In addition, many hydrophobic and Van der Waals interactions were regarded with Leu840, Val848, Ala866, Val899, Val916, Phe918, Leu1035, and Phe 1047. The molecular docking study revealed the ability of the tested compound to interact with the key amino acids in the ATP active site of VEGFR-2. The binding interactions and energy binding scores are in agreement with the obtained experimental in vitro anticancer and VEGFR-2 kinase inhibition activities for these compounds. 

#### 2.3.2. Physicochemical Properties and Drug-Likeness Predictions

The estimation of physicochemical parameters and drug-likeness profiles for the designed molecules is a crucial step for the development process of new drug candidates. Several active drug candidates were rejected during clinical trials because of their poor pharmacokinetics. Accordingly, in silico tools were developed to predict physicochemical parameters and drug-likeness profiles at an early stage of drug design to save time, reduce cost, and increase chances of success [71]. The oral bioavailability of the target compounds **3a**,**b**–**17a**,**b** were evaluated based on drug-likeness parameters of the Lipinski and the Veber rules.

The Lipinski rule of five is considered one of the highly influential rules in pharmacokinetic drug design. The Lipinski rule states that orally active molecules should have three of the following properties: molecular weight (MW) is less than 500, octanol/water partition coefficient (iLOG *P*) is less than five, and H-bond acceptors (HAs) and donors (HDs) are not more than ten and five, respectively [72]. The Veber rule states that compounds with a total polar surface area (TPSA) less than 140 Å^2^ and rotatable bonds (RBs) less than 10 were found to have better oral bioavailability [73].

The physicochemical properties of VEGFR-2 inhibitors **3a**,**b**–**17a**,**b** were calculated using the SwissADME online tool [74], and the key parameters are summarized in Table 7. 

The physicochemical properties data revealed that the MWs of the designed compounds were in the range from 266.25 to 465.27 g/mol, with **3a** as the lowest and **10g** as the highest, respectively. The iLOG *P* values varied between 0.78 and 2.93, with **8b** as lowest and **10g** as highest, respectively. The HDs and HAs of the molecules have been found in the range from 2 to 4 and 3 to 7, respectively. The number of RBs of the tested compounds was between 2 and 8. Compounds **7a**,**b** had the lowest TPSA with a value of 67.76 Å^2^, while compounds **10a** and **10c** had the highest TPSA with a value of 128.74 Å^2^. The obtained data demonstrated that all the target compounds **3a**,**b**–**17a**,**b** were in accordance with the Lipinski and Veber rules.

Additionally, two important pharmacokinetic parameters, passive GI absorption (HIA) and brain accessibility (BBB), of compounds **3a**,**b**–**17a**,**b** were predicted and graphically depicted by the Brain Or IntestinaL EstimateD permeation method (BOILED-Egg). The BOILED-Egg method predicts pharmacokinetic parameters by computing the lipophilicity (WLOG P) against the TPSA, as shown in Figure 11. The molecules located in the white region have a high probability of passive gastrointestinal absorption, while the molecules located in the yellow region (yolk) have a high probability of brain access. Moreover, the blue points represent molecules predicted to be actively effluxed by the P-glycoprotein (PGP+), while the red points represent molecules that are not predicted to be actively effluxed by the P-glycoprotein (PGP−) [75].

The results revealed that all the tested derivatives were suggested to have good intestinal absorption and bioavailability according to the Lipinski and Veber estimated parameters. Compounds **7a**,**b** were predicted to display high blood-brain permeability and gastrointestinal absorption. All the tested compounds, with the exception of **5a**,**b** and **8b**, were predicted to be not subjected to active efflux by P-glycoprotein (red dot). Compounds **5b**, **10e**, **10g**, **15a**, and **17a** could be potential candidates for drug discovery because of their potent cytotoxic activity and their good drug-likeness and pharmacokinetic properties.

## 3. Materials and Methods

### 3.1. Chemistry

All commercially purchased chemicals were used as received. Experiments were conducted under nitrogen or argon. Thin-layer chromatography (TLC) was performed on Merck TLC silica gel 60 F_254_ pre-coated aluminum sheets. Melting points were determined using Stuart electrothermal melting point apparatus and were uncorrected. Infrared spectra were recorded as KBr discs on a Thermo-912AO683 FT-IR spectrophotometer and were reported in frequency of absorption (cm^−1^). NMR spectra were recorded on a Bruker 400 MHz NMR spectrometer (Appendix A). Chemical shift (δ) values were reported in parts per million (ppm) relative to internal standard tetramethylsilane at δ 0.00 ppm. Coupling constants (*J*) were reported in Hertz. The spin multiplicity was abbreviated as: s, singlet; d, doublet; t, triplet; q, quartet; m, multiplet. Elemental analysis was performed on AnalysenSysteme GmbH-D-63452-HANAU apparatus and a Perkin Elmer 2400 CHN elemental analyzer. The values of elemental analyses were within ± 0.4% of the theoretical values.

#### 3.1.1. Synthesis of 4-((5-Substituted-2-oxoindolin-3-ylidene)amino)benzoic acid **3a**,**b**

Acids **3a**,**b** were prepared according to the general procedure described in [55].

#### 3.1.2. General Procedure for Synthesis of 2-(4-((5-Substituted-2-oxoindolin-3-ylidene)amino)benzoyl)-N-methylhydrazinecarbothioamide **5a**,**b**

To a mixture of benzoic acid, derivatives **3a**,**b** (3 mmol), EDC (3.3 mmol), 4-methylthiosemicarbazide **4** (3.6 mmol) and HOBt (3.9 mmol) in DMF (20 mL) were added to triethylamine (9 mmol) at 0 °C. The reaction mixture was stirred at 0 °C for one hour. Next, the ice bath was removed, and the reaction mixture was stirred at room temperature overnight. The reaction was quenched by addition of water (40 mL). The precipitated solid was filtered and purified by flash column chromatography using (hexanes/EtOAc, 3:1) as the mobile phase system to give N-methylthiosemicarbazises **5a**,**b**.

##### N-Methyl-2-(4-((2-oxoindolin-3-ylidene)amino)benzoyl)hydrazinecarbothioamide **5a**

Yellow powder (yield, 78 %); mp (°C) 249-251; FT-IR (KBr, cm^−1^): 3235 (NH), 1691(C=O), 1669 (C=O), 1587 (C=N); ^1^H NMR (400 MHz, DMSO) δ 12.61 (s, 2H), 11.22 (s, 1H), 9.25 (d, *J* = 4.5 Hz, 1H), 7.75–7.52 (m, 2H), 7.48–7.20 (m, 2H), 7.23–7.02 (m, 2H), 6.94 (d, *J* = 7.8 Hz, 2H), 3.10 (d, *J* = 4.6 Hz, 3H); ^13^C NMR (101 MHz, DMSO) δ 178.12, 163.10, 142.71, 132.05, 131.96, 131.65, 131.58, 122.80, 121.15, 121.04, 120.80, 120.56, 120.48, 111.54, 31.84; ESI-MS *m*/*z*: 354.3 [M+H]^+^; Anal. Calcd. for C_17_H_15_N_5_O_2_S: C, 57.78; H, 4.28; N, 19.82. Found: C, 57.62; H, 4.48; N, 19.95.

##### 2-(4-((5-Fluoro-2-oxoindolin-3-ylidene)amino)benzoyl)-N-methylhydrazinecarbothioamide **5b**

Orange powder (yield, 82 %); mp (°C) 252-254; FT-IR (KBr, cm^−1^): 3284 (NH), 1689 (C=O), 1654 (C=O), 1588 (CH=N); ^1^H NMR (400 MHz, DMSO) δ 12.48 (s, 1H), 11.21 (s, 2H), 9.31 (d, *J* = 4.5 Hz, 1H), 7.63–7.28 (m, 2H), 7.35–6.98 (m, 3H), 7.10–6.75 (m, 2H), 3.10 (d, *J* = 4.6 Hz, 3H); ^13^C NMR (101 MHz, DMSO) δ 178.09, 163.17, 159.87, 157.51, 138.94, 131.38 (d, *J* = 3.4 Hz), 121.95, 121.86, 117.88, 117.64, 112.62, 112.54, 108.16, 107.91, 31.78; ESI-MS *m*/*z*: 372.2 [M+H]^+^; Anal. Calcd. for C_17_H_14_FN_5_O_2_S: C, 54.89; H, 3.80; N, 18.86. Found: C, 55.05; H, 3.94; N, 18.97.

#### 3.1.3. Synthesis of Ethyl 4-((5-Substituted-2-oxoindolin-3-ylidene)amino)benzoate **7a**,**b**

Esters **7a**,**b** were prepared according to the general procedure described in [76].

#### 3.1.4. Synthesis of 4-((5-Substituted-2-oxoindolin-3-ylidene)amino)benzohydrazide **8a**,**b**

Hydrazides **8a**,**b** were prepared according to the general procedure described in [61].

#### 3.1.5. General Procedure for Synthesis of N′-(4-Substitutedbenzylidene)-4-((5-substituted-2-oxoindolin-3-ylidene)amino)benzohydrazide **10a**–**g**

Benzaldehyde derivatives **9a**–**g** (1.8 mmol) were added to a solution of hydrazides **8a**,**b** (1.8 mmol) in absolute ethanol (10 mL). Next, 0.3 mL of glacial acetic acid was added, and the reaction mixture was heated under reflux for 6 h. After cooling, the precipitate was collected, and the crude precipitate was recrystallized from methanol to give hydrazones **10a**–**g**. 

##### N′-(4-Nitrobenzylidene)-4-((2-oxoindolin-3-ylidene)amino)benzohydrazide **10a**

Orange powder (yield, 94%, *E*: *Z* = 5.3: 1); mp (°C) 220–222; FT-IR (KBr, cm^−1^): 3412 (NH), 1653 (C=O), 1636 (C=O), 1588 (C=N); ^1^H NMR (400 MHz, DMSO) δ 11.78 (s, 1H), 10.56–9.57 (d, *J* = 14.2 Hz, 1H), 8.49 (s, 1H), 8.29 (d, *J* = 8.8 Hz, 2H), 7.95 (d, *J* = 8.7 Hz, 2H), 7.71 (d, *J* = 8.6 Hz, 2H), 7.04–6.87 (m, 2H), 6.63 (d, *J* = 8.6 Hz, 2H), 5.91-5.86 (m, 2H); ^13^C NMR (101 MHz, DMSO) δ 164.67, 163.25, 156.87, 153.08, 147.99, 141.61, 139.11, 134.62, 130.17, 128.12, 126.69, 124.53, 122.73, 121.83, 119.46, 117.93, 113.09, 110.44; ESI-MS *m*/*z*: 414.5 [M+H]^+^; Anal. Calcd. for C_22_H_15_N_5_O_4_: C, 63.92; H, 3.66; N, 16.94. Found: C, 63.84; H, 3.89; N, 16.79.

##### N′-(4-Chlorobenzylidene)-4-((2-oxoindolin-3-ylidene)amino)benzohydrazide **10b**

White powder (yield, 89%, *E*: *Z* = 8.9: 1); mp (°C) 237–239; FT-IR (KBr, cm^−1^): 3398 (NH), 1666 (C=O), 1646 (C=O), 1596 (C=N); ^1^H NMR (400 MHz, DMSO) δ 11.55 (s, 1H), 8.40 (s, 1H), 7.82–7.51 (m, 3H), 7.51 (d, *J* = 8.5 Hz, 2H), 6.9–7.2 (m, 4H), 6.62 (d, *J* = 8.6 Hz, 2H), 5.90–5.72 (m, 2H); ^13^C NMR (101 MHz, DMSO) δ 163.35, 152.86, 144.95, 144.76, 134.52, 134.46, 134.25, 134.16, 129.93, 129.72, 129.52, 129.36, 129.20, 129.12, 128.91, 119.87, 119.80, 119.72, 113.19, 113.08; ESI-MS *m*/*z*: 403.9 [M+H]^+^; Anal. Calcd. for C_22_H_15_ClN_4_O_2_: C, 65.59; H, 3.75; N, 13.91. Found: C, 65.65; H, 3.97; N, 14.11.

##### N′-(3-Nitrobenzylidene)-4-((2-oxoindolin-3-ylidene)amino)benzohydrazide **10c**

Orange powder (yield, 93%, *E*: *Z* = 8:1); mp (°C) 200–201; FT-IR (KBr, cm^−1^): 3312 (NH), 1689 (C=O), 1664 (C=O), 1604 (C=N); ^1^H NMR (400 MHz, DMSO) δ 10.94 (s, 1H), 10.77–10.49 (m, 1H), 8.82–8.67 (m, 2H), 8.51–8.36 (m, 2H), 7.98–7.26 (m, 4H), 7.15 (t, *J* = 7.6 Hz, 1H), 7.14–6.73 (m, 4H); ^13^C NMR (101 MHz, DMSO) δ 164.72, 157.36, 150.49, 148.79, 145.73, 139.10, 135.47, 134.40, 131.29, 129.20, 126.58, 124.02, 122.93, 121.82, 117.92, 116.59, 111.48, 110.43; ESI-MS *m*/*z*: 414.3 [M+H]^+^; Anal. Calcd. for C_22_H_15_N_5_O_4_: C, 63.92; H, 3.66; N, 16.94. Found: C, 63.96; H, 3.71; N, 17.14.

##### N′-(4-Methylbenzylidene)-4-((2-oxoindolin-3-ylidene)amino)benzohydrazide **10d**

Reddish orange powder (yield, 84%, *E*: *Z* = 6.7:1); mp (°C) 234–236; FT-IR (KBr, cm^−1^): 3358 (NH), 1661 (C=O), 1636 (C=O), 1581 (C=N); ^1^H NMR (400 MHz, DMSO) δ 11.03 (s, 1H), 10.71–9.56 (m, 1H), 8.67 (s, 1H), 7.99–7.87 (m, 1H), 7.77 (d, *J* = 8.0 Hz, 2H), 7.65–7.46 (m, 1H), 7.42–7.33 (m, 3H), 7.16 (td, *J* = 7.7, 1.0 Hz, 2H), 7.02–6.91 (m, 3H), 2.37 (s, 3H); ^13^C NMR (101 MHz, DMSO) δ 165.08, 163.26, 161.73, 145.67, 141.80, 139.11, 134.88, 131.67, 130.31, 129.98, 128.79, 127.51, 126.69, 122.73, 121.83, 117.93, 116.27, 110.44, 21.63; ESI-MS *m*/*z*: 383.7 [M+H]^+^; Anal. Calcd. for C_23_H_18_N_4_O_2_: C, 72.24; H, 4.74; N, 14.65. Found: C, 72.33; H, 4.81; N, 14.55.

##### N′-(4-Methoxybenzylidene)-4-((2-oxoindolin-3-ylidene)amino)benzohydrazide **10e**

Orange powder (yield, 87%, *E*: *Z* = 5.7:1); mp (°C) 239–240; FT-IR (KBr, cm^−1^): 3339 (NH), 1683(C=O), 1659 (C=O), 1611 (C=N); ^1^H NMR (400 MHz, DMSO) δ 11.02 (s, 1H), 10.86 (s, 1H), 8.65 (d, *J* = 3.8 Hz, 2H), 7.98 (d, *J* = 8.8 Hz, 1H), 7.82 (d, *J* = 8.7 Hz, 2H), 7.43–7.38 (m, 1H), 7.15 (d, *J* = 8.8 Hz, 2H), 7.05 (dd, *J* = 10.3, 6.9 Hz, 4H), 6.93 (t, *J* = 7.3 Hz, 1H), 3.84 (s, 3H); ^13^C NMR (101 MHz, DMSO) δ 165.25, 163.12, 162.15, 160.99, 151.23, 145.68, 134.90, 133.96, 131.53, 130.46, 129.37, 128.68, 127.04, 123.04, 117.13, 115.27, 114.87, 111.60, 55.86; ESI-MS *m*/*z*: 399.2 [M+H]^+^; Anal. Calcd. for C_23_H_18_N_4_O_3_: C, 69.34; H, 4.55; N, 14.06. Found: C, 69.39; H, 4.47; N, 14.31.

##### N′-(4-(Dimethylamino)benzylidene)-4-((5-fluoro-2-oxoindolin-3-ylidene)amino)benzohydrazide **10f**

Reddish orange powder (yield, 91%, *E*: *Z* = 8:1); mp (°C) 202–204; FT-IR (KBr, cm^−1^): 3296 (NH), 1678 (C=O), 1643 (C=O), 1595 (C=N); ^1^H NMR (400 MHz, DMSO) δ 11.59 (s, 1H), 10.92 (m, 1H), 8.69 (s, 1H), 8.08–7.87 (m, 1H), 7.84 (d, *J* = 8.9 Hz, 1H), 7.44–7.30 (m, 1H), 7.30–7.18 (m, 1H), 7.18–7.04 (m, 1H), 7.07–6.93 (m, 1H), 6.95–6.77 (m, 4H), 6.81–6.68 (m, 1H), 3.07 (s, 3H), 2.92 (s, 3H); ^13^C NMR (101 MHz, DMSO) δ 167.11, 165.70, 163.55, 157.41, 153.86, 151.02, 141.13, 135.96 (d, *J* = 3.1 Hz), 132.01, 127.93, 124.92, 123.26, 120.61, 119.67, 114.67, 112.76, 112.36, 105.39, 40.45, 40.14; ESI-MS *m*/*z*: 430.6 [M+H]^+^; Anal. Calcd. for C_24_H_20_FN_5_O_2_: C, 67.12; H, 4.69; N, 16.31. Found: C, 67.37; H, 4.86; N, 16.52**.**

##### N′-(4-Bromobenzylidene)-4-((5-fluoro-2-oxoindolin-3-ylidene)amino)benzohydrazide **10g**

Orange powder (yield, 90%, *E*: *Z* = 7.3:1); mp (°C) 227–229; FT-IR (KBr, cm^−1^): 3312 (NH), 1675(C=O), 1657 (C=O), 1586 (C=N); ^1^H NMR (400 MHz, DMSO) δ 10.94 (s, 1H), 10.75–10.57 (m, 1H), 8.69 (s, 1H), 7.95 (d, *J* = 8.2 Hz, 3H), 7.83 (d, *J* = 6.8 Hz, 3H), 7.64 (d, *J* = 8.1 Hz, 1H), 7.32 (t, *J* = 8.9 Hz, 1H), 7.22–6.71 (m, 3H); ^13^C NMR (101 MHz, DMSO) δ 164.91, 163.46, 161.20, 159.13, 156.77, 150.87, 142.01, 132.88, 131.26 (d, *J* = 1.2 Hz), 126.57, 120.93, 120.70, 117.19, 117.10, 115.95, 115.70, 113.47, 112.52; ESI-MS *m*/*z*: 467.3 [M+H]^+^; Anal. Calcd. for C_22_H_14_FBrN_4_O_2_: C, 56.79; H, 3.03; N, 12.04. Found: C, 56.84; H, 3.11; N, 12.27.

#### 3.1.6. General Procedure for Synthesis of N-(1,3-dioxoisoindolin-2-yl)-4-((5-substituted-2-oxoindolin-3-ylidene)amino)benzamide **12a**,**b**

A mixture of hydrazide derivatives **8a**,**b** (2.1 mmol) and phthalic anhydride **11** (2.1 mmol) in 10 mL glacial acetic acid was stirred under sonication for 4 h at 50 °C. Next, the reaction mixture was cooled, and the reaction was quenched by addition of water (30 mL). The separated precipitate was collected and purified by flash column chromatography using (hexane/EtOAc, 4:1) as an eluent to afford cyclic imides **12a**,**b**.

##### N-(1,3-Dioxoisoindolin-2-yl)-4-((2-oxoindolin-3-ylidene)amino)benzamide **12a**

Orange powder (yield, 75%); mp (°C) 187–189; FT-IR (KBr, cm^−1^): 3365 (NH), 1696 (C=O), 1663 (C=O), 1596 (C=N); ^1^H NMR (400 MHz, DMSO) δ 12.84 (s, 1H), 11.25 (s, 1H), 8.43–7.85 (m, 1H), 7.86–7.61 (m, 2H), 7.70–7.46 (m, 2H), 7.47–7.25 (m, 2H), 7.25–6.77 (m, 5H); ^13^C NMR (101 MHz, DMSO) δ 173.14, 167.66, 162.99, 162.94, 142.74, 132.54, 131.83, 131.27, 127.49, 122.98, 121.81, 120.89, 120.31, 120.27, 117.91, 111.64, 111.54, 110.42; ESI-MS *m*/*z*: 411.2 [M+H]^+^; Anal. Calcd. for C_23_H_14_N_4_O_4_: C, 67.31; H, 3.44; N, 13.65. Found: C, 67.29; H, 3.70; N, 13.76. 

##### N-(1,3-Dioxoisoindolin-2-yl)-4-((5-fluoro-2-oxoindolin-3-ylidene)amino)benzamide **12b**

Orange powder (yield, 81%); mp (°C) 225–227; FT-IR (KBr, cm^−1^): 3342 (NH), 1689 (C=O), 1670 (C=O), 1612 (C=N); ^1^H NMR (400 MHz, DMSO) δ 10.73–10.65 (m, 1H), 9.81 (d, *J* = 14.8 Hz, 1H), 8.06–7.53 (m, 2H), 7.51–7.26 (m, 1H), 7.30–7.07 (m, 2H), 7.13–6.86 (m, 3H), 6.89–6.38 (m, 3H); ^13^C NMR (101 MHz, DMSO) δ 167.72, 163.47, 159.71, 157.37, 139.11, 135.23 (d, *J* = 1.4 Hz), 126.19, 126.16, 124.15, 124.06, 113.71, 113.47, 112.90, 111.30, 111.22, 105.08, 105.01, 104.75; ESI-MS *m*/*z*: 429.5 [M+H]^+^; Anal. Calcd. for C_23_H_13_FN_4_O_4_: C, 64.49; H, 3.06; N, 13.08. Found: C, 64.21; H, 2.89; N, 13.22.

#### 3.1.7. General Procedure for Synthesis of Ethyl 3-(2-(4-((5-Substituted-2-oxoindolin-3-ylidene)amino)benzoyl)hydrazono)butanoate **14a,b**

Ethyl acetoacetate **13** (2 mmol) was added to a solution of hydrazides **8a**,**b** (2 mmol) in absolute ethanol (10 mL). After the addition of 0.2 mL of glacial acetic acid, the reaction mixture was heated under reflux for 4 h. After cooling, the formed precipitate was collected, and the crude solid was recrystallized from methanol to afford hydrazones **14a**,**b**.

##### Ethyl 3-(2-(4-((2-oxoindolin-3-ylidene)amino)benzoyl)hydrazono)butanoate **14a**

Yellow powder (yield, 86%); mp (°C) 206–208; FT-IR (KBr, cm^−1^) 3321 (NH), 1685(C=O), 1673 (C=O), 1657 (C=O), 1614 (C=N); ^1^H NMR (400 MHz, DMSO) δ 14.19 (s, 1H), 11.03 (s, 1H), 7.48 (d, *J* = 7.2 Hz, 1H), 7.43–7.24 (m, 2H), 7.16 (t, *J* = 7.6 Hz, 1H), 7.12–6.96 (m, 2H), 6.95–6.81 (m, 2H), 5.07 (s, 2H), 4.13 (q, *J* = 6.8 Hz, 2H), 2.26 (s, 3H), 1.23 (t, *J* = 7.0 Hz, 3H); ^13^C NMR (101 MHz, DMSO) δ 167.61, 163.25, 162.26, 156.36, 141.77, 139.11, 132.36, 130.51, 127.50, 122.41, 121.10, 120.09, 117.92, 111.10, 110.43, 91.95, 59.49, 18.50, 14.85; ESI-MS *m*/*z*: 393.3 [M+H]^+^; Anal. Calcd. for C_21_H_20_N_4_O_4_: C, 64.28; H, 5.14; N, 14.28. Found: C, 63.96; H, 4.95; N, 13.95.

##### Ethyl 3-(2-(4-((5-Fluoro-2-oxoindolin-3-ylidene)amino)benzoyl)hydrazono)butanoate **14b**

Yellow powder (yield, 88%); mp (°C) 213–215; FT-IR (KBr, cm^−1^): 3354 (NH), 1698(C=O), 1675 (C=O), 1648 (C=O), 1598 (C=N); ^1^H NMR (400 MHz, DMSO) δ 14.20 (s, 1H), 11.04 (s, 1H), 7.37–7.13 (m, 2H), 7.16–7.13 (m, 2H), 7.11–6.73 (m, 3H), 5.11 (s, 2H), 4.13 (q, *J* = 7.1 Hz, 2H), 2.25 (s, 3H), 1.23 (t, *J* = 7.1 Hz, 3H); ^13^C NMR (101 MHz, DMSO) δ 167.56, 162.41, 159.77, 157.41, 156.20, 137.92, 131.83, 131.80 (d, *J* = 3.3 Hz), 122.43, 122.34, 116.80, 116.56, 112.02, 107.16, 106.91, 92.77, 59.60, 18.38, 14.82; ESI-MS *m*/*z*: 411.2 [M+H]^+^; Anal. Calcd. for C_21_H_19_FN_4_O_4_: C, 61.46; H, 4.67; N, 13.65. Found: C, 61.75; H, 4.88; N, 13.82.

#### 3.1.8. General Procedure for Synthesis of 5-Substituted-3-((4-(5-thioxo-4,5-dihydro-1,3,4-oxadiazol-2-yl)phenyl)imino)indolin-2-one **15a,b**

Potassium hydroxide (2 mmol) was added to a solution of hydrazides **8a**,**b** (2 mmol) and carbon disulfide (4 mmol) in 20 mL of absolute ethanol. The reaction mixture was refluxed for 12 h. Next, the reaction mixture was cooled, and the solvent was evaporated. The residue was dissolved in water and acidified with 10% HCl. The formed precipitate was collected and recrystallized from ethanol to generate the corresponding oxadiazole derivatives **15a**,**b**.

##### 3-((4-(5-Thioxo-4,5-dihydro-1,3,4-oxadiazol-2-yl)phenyl)imino)indolin-2-one **15a**

Yellow powder (yield, 86%); mp (°C) 219–220; FT-IR (KBr, cm^−1^): 3252 (NH), 1679(C=O), 1612 (C=N); ^1^H NMR (400 MHz, DMSO) δ 10.71 (s, 1H), 10.54 (s, 1H), 7.37 (d, *J* = 7.5 Hz, 2H), 7.30–7.10 (m, 2H), 7.08–6.77 (m, 4H); ^13^C NMR (101 MHz, DMSO) δ 186.12, 163.25, 160.80, 139.11, 127.52, 127.36, 126.67, 122.88, 122.73, 121.94, 121.84, 118.09, 117.93, 110.44; ESI-MS *m*/*z*: 323.5 [M+H]^+^; Anal. Calcd. for C_16_H_10_N_4_O_2_S: C, 59.62; H, 3.13; N, 17.38. Found: C, 59.68; H, 3.28; N, 17.51.

##### 5-Fluoro-3-((4-(5-thioxo-4,5-dihydro-1,3,4-oxadiazol-2-yl)phenyl)imino)indolin-2-one **15b**

Yellow powder (yield, 82%); mp (°C) 231–233; FT-IR (KBr, cm^−1^): 3263 (NH), 1668 (C=O), 1607 (C=N); ^1^H NMR (400 MHz, DMSO) δ 10.67 (d, *J* = 15.0 Hz, 2H), 7.17–7.14 (m, 1H), 7.08–6.91 (m, 3H), 6.91–6.73 (m, 3H); ^13^C NMR (101 MHz, DMSO) δ 186.84, 159.70, 157.36, 135.20 (d, *J* = 1.2 Hz), 135.19, 126.19, 126.16, 124.06, 113.69, 113.46, 111.29, 111.20, 105.01, 104.76; ESI-MS *m*/*z*: 341.3 [M+H]^+^; Anal. Calcd. for C_16_H_9_FN_4_O_2_S: C, 56.47; H, 2.67; N, 16.46. Found: C, 56.51; H, 2.70; N, 16.53.

#### 3.1.9. General Procedures for Synthesis of 2-(4-((5-Substituted-2-oxoindolin-3-ylidene)amino)benzoyl)-N-substitutedhydrazinecarbothioamide **17a**,**b**

Phenyl isothiocyanate **16** (1.9 mmol) was added to a solution of hydrazides **8a**,**b** (1.9 mmol) in absolute ethanol (10 mL). The reaction mixture was refluxed for 8 h. After cooling, the precipitated solid was collected and purified by flash column chromatography using (hexanes/EtOAc, 3:2) as an eluent to yield N-phenylthiosemicarbazides **17a**,**b**.

##### 2-(4-((2-Oxoindolin-3-ylidene)amino)benzoyl)-N-phenylhydrazinecarbothioamide **17a**

Yellow powder (yield, 93%); mp (°C) 211–213; FT-IR (KBr, cm^−1^): 3342 (NH), 1686 (C=O), 1654 (C=O), 1602 (C=N); ^1^H NMR (400 MHz, DMSO) δ 10.72 (s, 1H), 10.64–10.48 (m, 1H), 10.05 (d, *J* = 17.9 Hz), 9.57 (d, *J* = 13.4 Hz, 1H), 7.95 (d, *J* = 7.5 Hz, 1H), 7.89–7.60 (m, 2H), 7.56–7.30 (m, 6H), 7.17 (t, *J* = 7.3 Hz, 2H), 6.99 (t, *J* = 7.4 Hz, 1H), 6.88 (d, *J* = 7.6 Hz, 1H); ^13^C NMR (101 MHz, DMSO) δ 179.93, 163.27, 139.68, 139.12, 129.03, 128.92, 128.45, 127.54, 126.71, 126.17, 125.18, 124.92, 124.17, 122.74, 122.50, 121.86, 117.95, 110.46; ESI-MS *m*/*z*: 415.6 [M+H]^+^; Anal. Calcd. for C_22_H_17_N_5_O_2_S: C, 63.60; H, 4.12; N, 16.86. Found: C, 63.52; H, 4.29; N, 16.63.

##### 2-(4-((5-Fluoro-2-oxoindolin-3-ylidene)amino)benzoyl)-N-phenylhydrazinecarbothioamide **17b**

Orange powder (yield, 92%); mp (°C) 203–205; FT-IR (KBr, cm^−1^): 3339 (NH), 1694 (C=O), 1663 (C=O), 1593 (C=N); ^1^H NMR (400 MHz, DMSO) δ 12.70 (s, 1H), 11.14 (s, 1H), 10.68 (d, *J* = 15.0 Hz, 1H), 9.82 (d, *J* = 14.9 Hz, 1H), 7.67–7.61 (m, 2H), 7.45 (t, *J* = 7.7 Hz, 1H), 7.38–7.29 (m, 3H), 7.25–7.13 (m, 3H), 7.02–6.92 (m, 2H), 6.87–6.81 (m, 1H); ^13^C NMR (101 MHz, DMSO) δ 176.78, 163.48, 159.71, 157.37, 139.23, 138.78, 135.21 (d, *J* = 1.2 Hz), 128.94, 126.72, 126.20, 126.16, 124.06, 113.70, 113.46, 111.29, 111.21, 105.01, 104.76; ESI-MS *m*/*z*: 435.3 [M+H]^+^; Anal. Calcd. for C_22_H_16_FN_5_O_2_S: C, 60.96; H, 3.72; N, 16.16. Found: C, 60.69; H, 3.56; N, 16.33.

### 3.2. Biological Investigation

#### 3.2.1. In Vitro Cytotoxic Activity Assay

The anticancer activities of the target compounds **3a**,**b**–**17a**,**b** were quantitatively assessed using the MTT protocol against the MCF-7 and HepG2 cell lines, as described in the Appendix A. 

#### 3.2.2. In Vitro VEGFR-2 Kinase Inhibitory Assay

The most potent anti-proliferative derivatives, **5b**, **10e**, **10g**, **15a**, and **17a**, were tested for their VEGFR-2 kinase inhibition using a VEGFR2 (KDR) Kinase Assay Kit (Enzyme-Linked Immunosorbent Assay), as described in the Appendix A.

#### 3.2.3. Cell Cycle Analysis

The liver HepG2 cells were treated with 1.13 μM of **17a**, and the effect on the cell cycle distribution was evaluated by flow cytometric analysis, as described in the Appendix A.

#### 3.2.4. Apoptosis Analysis

The apoptotic ability of compound **17a** to the HepG2 cells was assessed using Annexin V-FITC/PI dual staining by flow cytometric analysis, as shown in the Appendix A.

#### 3.2.5. Caspase-3 and -9 Expression Assay

The expression of caspase-3 and -9 in the liver HepG2 cells treated with 1.13 μM of **17a** was determined using quantitative real-time PCR analysis, as presented in the Appendix A.

#### 3.2.6. BAX and Bcl-2 Expression Assay

The expression of apoptotic BAX and antiapoptotic Bcl-2 proteins in the liver HepG2 cells treated with 1.13 μM of **17a** was determined using Western blot analysis, as described in the Appendix A.

### 3.3. Molecular Modeling Studies

#### 3.3.1. Molecular Docking Study

The molecular docking simulation studies were performed on a Dell precision T3600 workstation with Intel Xeon^®^ CPU-1650.0 @ 3.20 GHz with Windows 7 operating system using Molecular Operating Environment software (MOE 2020.09) [70]; the docking protocol is described in the Appendix A.

#### 3.3.2. Physicochemical Properties and Drug Likeness Predictions

The physicochemical parameters of drug-likeness and the pharmacokinetic properties, such as gastrointestinal absorption, brain permeability, and P-glycoprotein efflux, were estimated using the SwissADME online tool (http://www.swissadme.ch/ access date 5 March 2022) for calculations [74].

## 4. Conclusions

In this investigation, a series of indoline-2-one derivatives **3a**,**b**–**17a**,**b** were designed and synthesized based on the pharmacophore model of the reported VEGFR-2 inhibitors. The results showed that compounds **5b**, **10e**, **10g**, **15a**, and **17a** exhibited the most potent anticancer activities with IC_50_ values from 0.74–4.62 µM against the breast MCF-7 cancer cell line (sunitinib IC_50_ = 4.77 µM) and from 1.13–8.80 µM against the liver HepG2 cancer cell line (sunitinib IC_50_ = 2.23 µM). Furthermore, these members displayed potent VEGFR-2 kinase inhibitory activities (IC_50_ from 0.078–0.358 µM) compared with sunitinib (IC_50_ = 0.139 µM). Moreover, the cell cycle of HepG2 cells was blocked by compound **17a** at the S phase, and the total apoptosis was enhanced by 24-fold. In addition, compound **17a** increased the expression of caspase-3, -9, and BAX by 6.88, 3.703, and 2.69-fold, and decreased the expression of Bcl-2 by 1.88 fold. The docking simulations demonstrated that the prepared compounds have similar interactions and orientation to sunitinib inside VEGFR-2. Finally, the molecular modeling studies showed that all the target compounds are not violating Lipinski and Veber rules for oral bioavailability and have promising drug-likeness behavior.

## Data Availability

Data is available within article and Appendix A.

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
