# Peer review of "Design, Synthesis, Molecular Modeling, and Anticancer Evaluation of New VEGFR-2 Inhibitors Based on the Indolin-2-One Scaffold"

_pharmaceuticals, 2022, doi:10.3390/ph15111416_

Round 1
Reviewer 1 Report
This manuscript by Abdelgawad et al. describes “Design, synthesis, molecular modeling, and anticancer evaluation of new VEGFR-2 inhibitors based on the indolin-2-one scaffold”. The introduction section, discussion, and conclusion in this manuscript is well written. Compounds were characterized with 1H NMR, 13C NMR, and elemental analysis. MTT assay results showed that some of these compounds displayed good cytotoxicity and potent VEGFR-2 kinase inhibitory activities in comparison to Sunitinib. I would recommend this manuscript to be published after as below.
1. The cytotoxicity of some of potent compounds should be assessed against at least one normal non-cancerous cell line.
2. The quality of figures 4 to 11 should be increased for clear visibility.
3. Most of compounds were obtained as mixture of E and Z isomer and authors has written data for major isomer. What was the major isomer and write the ratio of each other? Include the HPLC data for some potent compounds. In case of compounds containing fluorine, J value should be reported for 13C NMR.
Author Response
The authors would like to thank the editor and the reviewers for their valuable notes and fruitful discussion and suggestions which helped a lot to represent our work in the best version.
This manuscript by Abdelgawad et al. describes “Design, synthesis, molecular modeling, and anticancer evaluation of new VEGFR-2 inhibitors based on the indolin-2-one scaffold”. The introduction section, discussion, and conclusion in this manuscript is well written. Compounds were characterized with 1H NMR, 13C NMR, and elemental analysis. MTT assay results showed that some of these compounds displayed good cytotoxicity and potent VEGFR-2 kinase inhibitory activities in comparison to Sunitinib. I would recommend this manuscript to be published after as below.
- The cytotoxicity of some of potent compounds should be assessed against at least one normal non-cancerous cell line.
The main objective for the biological part of our study is to prove the activity of the new indoline-2-one series as antitumor and VEGFR-2 inhibitors. The biological studies were designed to prove this concept and offer data that could be used to prepare more potent derivatives in our project that will be tested later for safety on normal cells. So, the cytotoxicity study was performed on cancer cell lines to prove this concept. Similar publications are sharing the main objective with similar cytotoxicity studies as in:
Pharmaceuticals 2022, 15(10), 1262.
Bioorganic Chemistry 112 (2021) 104949
- The quality of figures 4 to 11 should be increased for clear visibility.
The quality of all figures has been enhanced and optimized. The previous upload in word file decreased the resolution of images. Individual photo files will be uploaded for publication quality.
- Most of compounds were obtained as mixture of E and Z isomer and authors has written data for major isomer. What was the major isomer and write the ratio of each other? Include the HPLC data for some potent compounds. In case of compounds containing fluorine, J value should be reported for 13C NMR
The major isomer was the E isomer. The ratio of E to Z isomers are added to compounds in the experimental section in the manuscript. The J values were also added to 13 C NMR of fluorine containing compounds also.
Thanks a lot
Best regards
Alaa M. Hayallah

Reviewer 2 Report
1. The cell cycle and apoptosis analysis studies are lacking positive control, please include a positive control.
2. The elemental analysis only covers C, H, and N atoms. Please add the MS and HRMS results for the molecular weights of the resulting compounds.
3. Please avoid inappropriate self-citations and only cite the ones that are related to the current study.
4. Please adequately described the methods with more details in the supporting information. For example, what solvent was used to prepare the compound solution, what concentration of VEGFR-2 was used, and so on?
5. For the docking study, please enlarge the size and increase the resolution of 3D presentations. In addition, please indicate the dipole/arene cation interactions in the 3D presentation as well.
6. Please briefly explain why you choose MCF-7 and HepG2 for the study.
7. Type II inhibitor targets inactive VEGFR-2, and binds to both ATP catalytic site and the adjacent hydrophobic pocket.
8. On Page 16 lines 11-16, the drug-likeness are predicted results, the current description is too affirmative.
9. Please add the line number, and revise the current manuscript as well as the supporting information thoroughly. There are many typos, for example, on page 14, the figure legends, and line 1.
Author Response
Dear reviewer
The authors would like to thank the editor and the reviewers for their valuable notes and fruitful discussion and suggestions which helped a lot to represent our work in the best version.
- The cell cycle and apoptosis analysis studies are lacking positive control, please include a positive control.
The cell cycle and apoptosis analysis are generally carried out to explore the ability to induce cell cycle arrest and programmed cell death. Most similar studies on different publications are comparing their obtained data with a negative control to elucidate that effect. Reported data for sorafenib were added and discussed in the updated manuscript.
- The elemental analysis only covers C, H, and N atoms. Please add the MS and HRMS results for the molecular weights of the resulting compounds.
The MS data are added to the updated manuscript in the experimental part
- Please avoid inappropriate self-citations and only cite the ones that are related to the current study.
All references are updated and checked as reviewer asked.
- Please adequately described the methods with more details in the supporting information. For example, what solvent was used to prepare the compound solution, what concentration of VEGFR-2 was used, and so on?
The supporting information methods were updated to include comprehensive details.
- For the docking study, please enlarge the size and increase the resolution of 3D presentations. In addition, please indicate the dipole/arene cation interactions in the 3D presentation as well.
The quality of all figures has been enhanced and optimized. The word file decreased the resolution of images. Individual photo files will be uploaded for publication quality. The dipole arene interaction is not default shown in 3d representation in the MOE version used.
- Please briefly explain why you choose MCF-7 and HepG2 for the study.
The two cell line were chosen due to the high expression of VEGF/ VEGFR-2 in these cancerous cells which represents a good target to test the efficacy of the designed VEGFR-2 inhibitors.
Kelly J. Higgins, Shengxi Liu, Maen Abdelrahim, Kathryn Vanderlaag, Xinyi Liu, Weston Porter, Richard Metz, Stephen Safe, Molecular Endocrinology, Volume 22, Issue 2, 1 February 2008, Pages 388–402.
A-RANG LEE, SU-MIN BAEK, SEOUNG-WOO LEE, TAE-UN KIM, JEE EUN HAN, SEULGI BAE, SANG-JOON PARK, TAE-HWAN KIM, KYU-SHIK JEONG, SEONG-KYOON CHOI and JIN-KYU PARK, In Vivo. 2021 May-Jun; 35(3): 1473–1483.
- Type II inhibitor targets inactive VEGFR-2, and binds to both ATP catalytic site and the adjacent hydrophobic pocket.
The text is updated and addressed.
- On Page 16 lines 11-16, the drug-likeness are predicted results, the current description is too affirmative.
The language was rewritten to emphasize that the drug-likeness properties are predicted data.
- Please add the line number, and revise the current manuscript as well as the supporting information thoroughly. There are many typos, for example, on page 14, the figure legends, and line 1.
The manuscript was checked for the typo errors and all suggestions are addressed.
Thanks a lot
Best regards
Alaa H. Hayallah

Reviewer 3 Report
The article “Design, synthesis, molecular modeling, and anticancer evaluation of new VEGFR-2 inhibitors based on the indolin-2-one scaffold” describes a new series of 23 indoline-2-one derivatives with antiproliferative activity against breast and liver cell lines, due to their VEGFR-2 inhibition activity. Synthesis, in vitro screening and molecular modelling studies are clearly presented, comparing the best candidate (compound 17a) with the FDA-approved drug Sunitinib. Results are relevant in the field, while material and methods are properly organized. Hence, it is suitable for publications after the following minor revisions:
- In figure 3, compounds 3a,b are missing;
- At page 8, after the first period, insert the IC50 range;
- Authors should report the cytotoxic activity against healthy cell lines;
- In table 2, the columns 2, 3 and 4 are redundant. Please keep the info about the structures in table 1 only;
- In paragraph 2.2.3, in analogy with the previous assays, compare the cell cycle results with those from other VEGFR-2 inhibitors (for example Sunitinib);
- Merge figures 8, 9 and 10 to easily compare the 2D and 3D representations of the ligands.
- At page 1, add more recent and appropriate citations when describing “the battle to develop potent and safe chemotherapeutic agents” to prove the concrete efforts of medicinal chemists in the field: Drug Dev Res. 2022;83:1331–1341 DOI: 10.1002/ddr.21962; Molecules. 2022 Oct 10;27(19):6758. doi:10.3390/molecules27196758; Eur J Med Chem. 237:114399 DOI: 10.1016/j.ejmech.2022.114399; Eur J Med Chem. 2022 Aug 5;238:114422. doi: 10.1016/j.ejmech.2022.114422; ACS Med. Chem. Lett. 2022, 13, 3, 358 - 364 https://doi.org/10.1021/acsmedchemlett.1c00600;
- The are some typos in the text. For example, at pg 7, compounds 3a,b – 17a,b should be in bold; in the caption of figure 10, change 17b with 17a
Author Response
Dear Reviewer
The authors would like to thank the editor and the reviewers for their valuable notes and fruitful discussion and suggestions which helped a lot to represent our work in the best version.
Comments and Suggestions for Authors
The article “Design, synthesis, molecular modeling, and anticancer evaluation of new VEGFR-2 inhibitors based on the indolin-2-one scaffold” describes a new series of 23 indoline-2-one derivatives with antiproliferative activity against breast and liver cell lines, due to their VEGFR-2 inhibition activity. Synthesis, in vitro screening and molecular modelling studies are clearly presented, comparing the best candidate (compound 17a) with the FDA-approved drug Sunitinib. Results are relevant in the field, while material and methods are properly organized. Hence, it is suitable for publications after the following minor revisions:
- In figure 3, compounds 3a,b are missing;
Compounds 3a, b were added to figure 3
- At page 8, after the first period, insert the IC50 range;
The manuscript was checked for the typo errors and all suggestions are addressed.
- Authors should report the cytotoxic activity against healthy cell lines;
The main objective for the biological part of our study is to prove the activity of the new indoline-2-one series as antitumors and VEGFR-2 inhibitors. The biological studies were designed to prove this concept and offer data that could be used to prepare more potent derivatives that will be tested later for safety on normal cells. So, the cytotoxicity study was performed on cancer cell lines to prove this concept. Similar publications are sharing the main objective with similar cytotoxicity studies as in:
Pharmaceuticals 2022, 15(10), 1262.
Bioorganic Chemistry 112 (2021) 104949
- In table 2, the columns 2, 3 and 4 are redundant. Please keep the info about the structures in table 1 only;
The tables are updated and addressed as advised.
- In paragraph 2.2.3, in analogy with the previous assays, compare the cell cycle results with those from other VEGFR-2 inhibitors (for example Sunitinib);
The suggestion was addressed and the data were compared to sorafenib, VEGFR-2 inhibitor.
- Merge figures 8, 9 and 10 to easily compare the 2D and 3D representations of the ligands.
The size of the merged file was large and the resolution decreased, individual photo files are uploaded for high quality.
- At page 1, add more recent and appropriate citations when describing “the battle to develop potent and safe chemotherapeutic agents” to prove the concrete efforts of medicinal chemists in the field: Drug Dev Res. 2022;83:1331–1341 DOI: 10.1002/ddr.21962; Molecules. 2022 Oct 10;27(19):6758. doi:10.3390/molecules27196758; Eur J Med Chem. 237:114399 DOI: 10.1016/j.ejmech.2022.114399; Eur J Med Chem. 2022 Aug 5;238:114422. doi: 10.1016/j.ejmech.2022.114422; ACS Med. Chem. Lett. 2022, 13, 3, 358 - 364 https://doi.org/10.1021/acsmedchemlett.1c00600;
All the advised references are added to the updated manuscript.
- The are some typos in the text. For example, at pg 7, compounds 3a,b – 17a,b should be in bold; in the caption of figure 10, change 17b with 17a
The manuscript was checked for the typo errors and all suggestions are addressed.
Tahnks a lot
Best regards
Alaa M. Hayallah

Round 2
Reviewer 1 Report
Authors have provided answers to comments and corrected the manuscript. I would recommend this may be considered for publication in Pharmaceuticals journal.
Reviewer 2 Report
After reviewing the current manuscript, my recommendation is 'Accept in present form'.